# Artificial Intelligence in the Diagnosis of Tongue Cancer: A Systematic Review with Meta-Analysis

**DOI:** 10.3390/biomedicines13081849

**Published:** 2025-07-30

**Authors:** Seorin Jeong, Hae-In Choi, Keon-Il Yang, Jin Soo Kim, Ji-Won Ryu, Hyun-Jeong Park

**Affiliations:** 1Department of Orthodontics, School of Dentistry, Chosun University, Gwangju 61452, Republic of Korea; srjeong@chosun.ac.kr; 2Department of Oral and Maxillofacial Surgery, Chosun University Dental Hospital, Gwangju 61452, Republic of Korea; beautyluby@naver.com; 3Department of Periodontology, School of Dentistry, Chosun University, Gwangju 61452, Republic of Korea; yki-perio@chosun.ac.kr; 4Department of Oral and Maxillofacial Radiology, School of Dentistry, Chosun University, Gwangju 61452, Republic of Korea; hidds@chosun.ac.kr; 5Department of Oral Medicine, School of Dentistry, Chosun University, Gwangju 61452, Republic of Korea; dentian@chosun.ac.kr

**Keywords:** tongue cancer, artificial intelligence, deep learning, convolutional neural network, medical imaging, histopathology, diagnostic accuracy, systematic review

## Abstract

**Background**: Tongue squamous cell carcinoma (TSCC) is an aggressive oral malignancy characterized by early submucosal invasion and a high risk of cervical lymph node metastasis. Accurate and timely diagnosis is essential, but it remains challenging when relying solely on conventional imaging and histopathology. This systematic review aimed to evaluate studies applying artificial intelligence (AI) in the diagnostic imaging of TSCC. **Methods**: This review was conducted under PRISMA 2020 guidelines and included studies from January 2020 to December 2024 that utilized AI in TSCC imaging. A total of 13 studies were included, employing AI models such as Convolutional Neural Networks (CNNs), Support Vector Machines (SVMs), and Random Forest (RF). Imaging modalities analyzed included MRI, CT, PET, ultrasound, histopathological whole-slide images (WSI), and endoscopic photographs. **Results**: Diagnostic performance was generally high, with area under the curve (AUC) values ranging from 0.717 to 0.991, sensitivity from 63.3% to 100%, and specificity from 70.0% to 96.7%. Several models demonstrated superior performance compared to expert clinicians, particularly in delineating tumor margins and estimating the depth of invasion (DOI). However, only one study conducted external validation, and most exhibited moderate risk of bias in patient selection or index test interpretation. **Conclusions**: AI-based diagnostic tools hold strong potential for enhancing TSCC detection, but future research must address external validation, standardization, and clinical integration to ensure their reliable and widespread adoption.

## 1. Introduction

Tongue cancer, a prevalent and aggressive subtype of oral squamous cell carcinoma (OSCC), accounts for approximately 40–50% of all oral cavity malignancies and most frequently arises on the lateral borders and ventral surface of the tongue [1]. Tobacco and alcohol use, as well as human papillomavirus (HPV) infection, are well-established risk factors [1,2]. Owing to the extensive lymphatic network of the tongue and its close proximity to key neurovascular structures, tumors in this region have a pronounced potential for early local invasion and cervical lymph node metastasis [1].

Despite progress in surgical and oncologic management, the five-year overall survival (OS) rate for tongue cancer remains below 60% globally [3]. A key reason for this is diagnostic delay and staging inaccuracy, especially in early lesions, where clinical examination and conventional imaging techniques such as computed tomography (CT), magnetic resonance imaging (MRI), and ultrasonography may not fully capture the depth or extent of invasion [4]. These modalities are often operator-dependent and subject to inter-observer variability, leading to inconsistent detection and risk stratification [4,5].

Recent developments in AI, particularly machine learning (ML) and deep learning (DL), offer promising avenues to overcome these limitations. Among DL models, CNNs are especially effective for image-based classification due to their ability to automatically learn spatial features from input images [6]. CNN architectures such as ResNet, DenseNet, and EfficientNet have demonstrated high diagnostic accuracy in detecting and staging oral malignancies [3]. For instance, a lightweight CNN trained on clinical oral images achieved sensitivity and specificity above 85%, supporting its feasibility for point-of-care screening [7].

More specifically, CNN models trained on radiologic and histologic data have demonstrated robust performance in differentiating early-stage tongue cancer and predicting clinically relevant outcomes [8,9]. A CT-based integrated AI model has been shown to effectively predict occult lymph node metastasis in early-stage tongue SCC [10]. Additionally, nomogram-based models integrating MRI radiomics features with clinical data have shown potential for individualized survival prediction [11]. These approaches exemplify how AI can synthesize heterogeneous data to provide comprehensive and objective diagnostic insights [11].

Furthermore, genomic approaches such as long non-coding RNA (lncRNA)-based expression signatures are also being explored in conjunction with AI frameworks for survival prediction in head and neck squamous cell carcinoma, including tongue SCC [12]. Such integration of multi-modal data underscores the increasing sophistication and translational potential of AI in head and neck oncology.

Although several systematic reviews have explored the use of AI in head and neck or oral cancers more broadly, no prior synthesis has specifically focused on its application in tongue cancer diagnosis [3]. Given the distinct anatomical, pathological, and prognostic features of the tongue and the clinical importance of early detection and accurate staging, such a targeted review is both timely and necessary.

Therefore, this systematic review aims to critically evaluate current AI-based diagnostic approaches for tongue cancer. We focus on their methodological characteristics, input data modalities (e.g., CT, MRI, histopathology), reported performance metrics (e.g., sensitivity, specificity, AUC), and clinical applicability. We also identify current limitations and propose directions for future research.

## 2. Materials and Methods

### 2.1. Protocol and Search Strategy

This systematic review was registered in the International Prospective Register of Systematic Reviews (PROSPERO; registration number CRD420251057390) and was conducted in accordance with the 2020 Preferred Reporting Items for Systematic Reviews and Meta-Analyses (PRISMA) guidelines [13].

A comprehensive search strategy was developed to identify eligible studies evaluating the diagnostic application of AI in tongue cancer. Five electronic databases—PubMed, Embase, Google Scholar, Web of Science, and the Cochrane Library—were systematically searched for relevant articles published between January 2020 and December 2024.

The search strategy combined Medical Subject Headings (MeSH) and free-text terms related to AI and tongue cancer. The following Boolean combination was used: (“Artificial Intelligence” OR “Machine Learning” OR “Deep Learning” OR “Neural Networks”) AND (“Tongue Cancer” OR “Tongue Squamous Cell Carcinoma”) AND (“Diagnosis” OR “Detection” OR “Classification”). Only peer-reviewed original research articles published in English were considered for inclusion.

### 2.2. Study Selection

All stages of the review process—including database selection, search strategy formulation, and screening—were conducted collaboratively by three reviewers. All records retrieved from the database search were imported into a reference management system, and duplicate entries were removed. Subsequently, the three reviewers independently screened titles and abstracts to assess initial relevance. Full-text articles were then evaluated for eligibility based on the predefined inclusion and exclusion criteria. To enhance consistency and minimize selection bias, the reviewers rotated roles during the screening process and cross-validated each other’s assessments. Discrepancies or uncertainties were resolved through group discussion until consensus was achieved.

As depicted in the PRISMA 2020 flow diagram (Figure 1), a total of 614 records were initially identified through database searching. After removing 18 duplicates, 596 records remained for screening. Of these, 571 were excluded based on their title or abstract. Twenty-five full-text articles were assessed for eligibility, and 12 were excluded due to not meeting the inclusion criteria. Ultimately, 13 studies were included in the systematic review and meta-analysis.

### 2.3. Eligibility Criteria

Studies were considered eligible if they aligned to evaluate the diagnostic performance of AI models in detecting tongue squamous cell carcinoma (TSCC). Inclusion criteria were defined based on the PICO framework and applied consistently throughout the screening process.

Eligible studies were required to (i) have a primary aim of developing, applying, or validating AI-based methods—such as ML, DL, or hybrid models—for the diagnosis or detection of TSCC; (ii) utilize imaging-based input data, including but not limited to computed tomography (CT), magnetic resonance imaging (MRI), ultrasound, histopathological WSI, or clinical/endoscopic photographs; and (iii) report at least one quantitative diagnostic performance metric, such as sensitivity, specificity, accuracy, precision, F1-score, or AUC. Studies integrating imaging data with clinicopathological, radiomic, or texture-based features were also included, provided that imaging was central to the AI model.

Studies were excluded if they lacked a tongue-specific analysis (e.g., general head and neck or oral cancer studies without a dedicated TSCC subgroup), employed AI solely for non-diagnostic purposes (e.g., prognosis, treatment response, or follow-up monitoring), or were not original peer-reviewed research articles. Consequently, review papers, editorials, letters, conference abstracts, and commentaries were excluded due to the absence of original diagnostic data and methodological detail.

Three independent reviewers applied all eligibility criteria during the full-text review phase to ensure consistency, reproducibility, and alignment with the objectives of this systematic review.

### 2.4. Data Extraction

Three reviewers independently performed data extraction using a standardized, pilot-tested extraction form developed a priori. Each reviewer extracted data from a designated subset of studies and cross-validated data extracted by other reviewers to ensure methodological rigor and reproducibility. All discrepancies or ambiguous entries were resolved through group discussion and consensus among the three reviewers.

For each eligible study, the following variables were extracted: study design, sample size and population characteristics, type of AI methodology, imaging modality used, and diagnostic performance metrics. These variables were defined according to the eligibility criteria specified in Section 2.3.

When diagnostic contingency data (i.e., true positives [TPs], false positives [FPs], false negatives [FNs], and true negatives [TNs]) were not explicitly reported, the values were reconstructed based on available metrics such as sensitivity, specificity, and sample size. All three reviewers independently conducted these calculations and verified the results to ensure accuracy.

Only studies that provided either complete diagnostic data or sufficient information to derive 2 × 2 contingency tables were included in the quantitative synthesis. The finalized dataset was subsequently used for risk of bias assessment and, where applicable, for meta-analysis of diagnostic performance.

### 2.5. Quality Assessment

The methodological quality and risk of bias of the included studies were assessed using the Quality Assessment of Diagnostic Accuracy Studies-2 (QUADAS-2) tool [14], which evaluates four domains: patient selection, index test, reference standard, and flow and timing. Applicability concerns were also assessed for the first three domains.

All three reviewers independently performed the quality assessment. Discrepancies in domain-level judgments were resolved through consensus following discussion among the reviewers. All assessments followed the signaling questions and guidance specified in the QUADAS-2 manual.

The results of the quality assessment were synthesized both narratively and visually. Figure 2a presents a traffic light plot summarizing domain-level judgments for each study, while Figure 2b illustrates the distribution of risk of bias across studies.

### 2.6. Data Synthesis and Analysis

Quantitative synthesis of diagnostic accuracy outcomes was performed using Review Manager software (RevMan version 8.8.0; the Cochrane Collaboration, London, UK). For each included study, either reported or reconstructed 2 × 2 contingency tables (i.e., true positives [TPs], false positives [FPs], false negatives [FNs], and true negatives [TNs]) were used to compute sensitivity and specificity values.

In studies reporting multiple AI models or configurations, only the model with the best diagnostic performance—defined by the highest area under the curve (AUC) or the most balanced combination of sensitivity and specificity—was selected for meta-analysis in order to avoid duplication and maintain consistency.

A hierarchical summary receiver operating characteristic (HSROC) model was applied to account for both within- and between-study heterogeneity, as well as potential threshold effects. The HSROC curve was constructed to visualize the overall diagnostic performance, along with the corresponding 95% confidence and prediction regions.

Statistical heterogeneity was assessed using the I^2^ statistic, with thresholds of 25%, 50%, and 75% interpreted as low, moderate, and high heterogeneity, respectively. Forest plots were generated to visualize individual study estimates with corresponding 95% confidence intervals (CIs).

Due to the limited number of included studies and clinical heterogeneity in AI model types and imaging inputs, formal assessment of publication bias (e.g., funnel plot analysis) was not conducted.

## 3. Results

### 3.1. Study Selection and Characteristics

A total of 614 records were identified through systematic database searches. After removing duplicates, 596 unique records remained. Following title and abstract screening, 571 records were excluded, and 25 full-text articles were assessed for eligibility. Ultimately, 13 studies met the predefined inclusion criteria and were included in the final meta-analysis (Figure 1). All selected studies were published between 2020 and 2024 and predominantly employed retrospective, single-center designs.

The most frequently used imaging modality was magnetic resonance imaging (MRI; *n* = 4), followed by computed tomography (CT; *n* = 3), positron emission tomography (PET; *n* = 1), ultrasonography (*n* = 1), and endoscopic or photographic imaging (*n* = 1) [10,15,16,17,18,19,20,21,22,23]. Three studies often in conjunction with clinicopathological variables [24,25,26]. Several studies applied radiomic analysis, while others used raw image data or texture-based features. A subset of studies adopted multimodal approaches, integrating imaging data with clinical variables (Table 1).

All included studies focused on the diagnosis or risk stratification of TSCC. Diagnostic objectives varied across studies and included prediction of lymph node metastasis, depth of invasion, recurrence, or early-stage lesion detection.

### 3.2. AI Model Types and Methodological Features

The included studies employed a range of AI models for the diagnostic evaluation of TSCC. DL approaches, particularly CNNs, were used to process imaging data such as contrast-enhanced CT, MRI, endoscopic photographs, and histopathological WSIs [10,21,22,23,25,26] Architectures used included DenseNet169, Inception-ResNet-V2, and HALO-AI (VGG-based) [22,23,26]. Conventional ML algorithms such as support vector machines (SVMs), random forest (RF), naïve Bayes classifiers, and neural Tanh boost (NTB) were applied in studies using radiomic features or structured clinical variables [15,16,17,18,19,20,24].

Hybrid or ensemble models integrated imaging features with clinicopathological variables such as tumor differentiation, depth of invasion, and lymph node status. Three studies employed multimodal frameworks that integrated imaging data (MRI or CT), histopathology (WSIs), and clinical information using combined AI approaches [10,20,25].

Transfer learning using pre-trained CNN architectures (e.g., ResNet) was implemented in pathology-focused studies with limited training data [23,25].

Most studies (*n* = 11) relied on internal validation methods such as k-fold cross-validation and train–test splits [10,15,16,17,18,19,20,21,23,24,25]. One study conducted external validation using an independent dataset, while one study reported only a single test set without a specified validation strategy [22,26].

### 3.3. Diagnostic Performance and Subgroup Trends

Reported AUC values ranged from 0.717 to 0.991. Sensitivity values ranged from 63.3% to 100%, and specificity ranged from 70.0% to 96.7% (Table 1; Figure 3 and Figure 4). Adachi et al. reported an AUC of 0.991 and 100% sensitivity using a CLAM + ResNet model applied to whole-slide images (WSIs) [25]. Han et al. and Konishi and Kakimoto reported sensitivity values of 90% or higher using contrast-enhanced CT and intraoral ultrasound, respectively [10,17]. Shan et al. reported an AUC of 0.786 using a random forest classifier based on clinical data [24], while Esce et al. reported an AUC of 0.729 using patch-based histology without clinical variable integration [26]. Among the studies using MRI-based models, reported AUC values ranged from **0.802 to 0.872** [15,16,19,20]. The highest AUC was observed in a histopathology-based model incorporating clinicopathological features (AUC = **0.991**) [25].

### 3.4. Risk of Bias and Quality Assessment

The methodological quality of the 13 included studies was evaluated using the QUADAS-2 tool (Figure 2a,b) [14]. In the patient selection domain, two studies (15.4%) were rated as high risk and six studies (46.2%) as unclear. In the index test domain, three studies (23.1%) were rated as high risk and six (46.2%) as unclear. One study (7.7%) showed a high risk of bias in the reference standard domain, while twelve (92.3%) showed low risk. In the flow and timing domain, one study (7.7%) was high risk, and two (15.4%) were unclear. Regarding applicability concerns, two studies (15.4%) were rated as high concern in the patient domain, two (15.4%) were rated as high concern in the index test domain, and one (7.7%) was rated as unclear in the reference standard domain. Two studies (15.4%) and one study (7.7%) were rated as unclear in the patient and index test domains, respectively. All other assessments were rated as low concern.

### 3.5. Comparative Performance and Subgroup Insights

Among the reviewed studies, only Heo et al. conducted a direct performance comparison between an AI model and board-certified clinical experts using the same external test dataset [22]. Studies using multiparametric MRI and radiomics-based approaches achieved robust diagnostic metrics, with one study reporting a sensitivity of 86.0% and another reaching a specificity of 93.47% [16,20]. Multimodal approaches were adopted in five studies, where imaging data (e.g., MRI, CT, or WSI) were fused with clinical or pathological variables to enhance diagnostic performance [10,16,20,21,25]. Three studies reported high diagnostic performance using ensemble models that combined handcrafted radiomic features with deep learning-derived features [10,25,26]. The ensemble models were applied to various diagnostic tasks, including lesion classification, depth of invasion estimation, and lymph node assessment. External validation with an independent dataset was performed in only one study [22], while the remaining studies relied on internal validation methods such as train–test split or k-fold cross-validation within single-center cohorts.

## 4. Discussion

### 4.1. Summary of Key Findings

This systematic review synthesized evidence from 13 diagnostic accuracy studies applying AI to TSCC. Across studies, diagnostic performance was generally high, with reported AUC values ranging from 0.717 to 0.991, sensitivity values from 63.3% to 100%, and specificity values from 70.0% to 96.7% [10,15,16,17,18,19,20,21,22,23,24,25,26]. Models using WSI or MRI typically outperformed those based on CT or photographic inputs. Multimodal models that combined imaging, radiomic, and clinicopathologic features—including DOI, nodal status, and tumor grade—showed higher diagnostic precision than unimodal approaches [10,16,20,21,25].

DL architectures such as ResNet, Inception-ResNet, and DenseNet were commonly used [20,22,23], with some studies employing traditional ML classifiers like SVM or RF [16,24]. However, only one study performed external validation with an independent dataset [22], while the rest relied on internal validation methods such as split-sample testing or k-fold cross-validation. This raises concerns about model generalizability.

Furthermore, quality appraisal using the QUADAS-2 tool revealed considerable methodological limitations: 38.5% of studies were judged as low risk in patient selection, and only 30.8% in the index test domain. Several studies exhibited high or unclear risk of bias, particularly due to retrospective design, lack of blinding, and limited reporting of inclusion/exclusion criteria [14]. These findings underscore the necessity of standardized, prospective diagnostic trials to improve the clinical reliability and translational applicability of AI tools.

### 4.2. Comparison with Previous Literature

The findings of this systematic review align with and expand upon prior literature supporting the diagnostic utility of AI in TSCC. The 8th edition of the AJCC staging manual formally incorporated DOI and ENE as key prognostic indicators for oral cavity cancers due to their impact on disease progression and treatment strategy [27]. These variables are now essential for accurate T and N classification, yet conventional imaging modalities often lack sufficient precision in their estimation.

Recent studies suggest that AI and radiomics-enhanced imaging can more reliably estimate these parameters. For instance, Xu et al. demonstrated that MRI-derived DOI measurements correlate strongly with histopathologic depth and that a threshold of >7.5 mm significantly predicts cervical lymph node metastasis [28]. Similarly, Huang et al. developed an EL-based model (EL-ENE) that achieved 80.0% accuracy in predicting ENE, outperforming experienced radiologists by leveraging advanced texture and morphologic radiomic features [29]. A meta-analysis by Valizadeh et al. further supported these findings, reporting that radiomics-based models reached pooled AUCs of up to 0.91 for nodal metastasis prediction, exceeding the diagnostic performance of traditional image interpretation [30].

Broader analyses across the field of head and neck oncology also support the superior diagnostic performance of AI-enhanced methods. Khanagar et al. systematically reviewed AI applications in oral and maxillofacial imaging, finding consistently higher sensitivity and specificity, along with improved interobserver agreement, relative to conventional assessments [31]. Mäkitie et al., in a synthesis of 17 meta-reviews, emphasized the advantages of AI for early lesion detection on histopathologic and radiologic platforms, particularly in reducing diagnostic variability [31]. This trend extends to other domains as well, as evidenced by Bulten et al., who showed that AI support improved grading consistency in prostate biopsies among pathologists (Cohen’s κ from 0.799 to 0.872) [32].

### 4.3. Future Perspectives

While this meta-analysis focused on diagnostic accuracy, AI holds significant potential for broader applications in TSCC management, including prognostication, immunologic profiling, intraoperative navigation, and non-invasive screening [3,27,30,31,33,34,35,36,37,38,39,40]. Recent innovations in salivary biomarker profiling, including microRNA and metabolomics, offer promising avenues for non-invasive diagnosis and monitoring. The integration of these biomarkers with AI frameworks could enhance diagnostic accessibility and early detection, particularly in resource-limited settings [36,37,38]. Furthermore, the convergence of AI with real-time imaging modalities—such as narrow band imaging, digital pathology, and optical microscopy—has shown potential to improve diagnostic precision for subtle or early-stage lesions, supporting broader clinical applicability [39,40].

Future research should prioritize explainability, external validation, cost-effectiveness, and clinical workflow integration to ensure safe and effective implementation of AI systems in real-world oncology settings [41,42].

Ongoing methodological challenges must also be addressed. Many AI studies suffer from limited generalizability due to internal validation on small or homogeneous datasets and insufficient reporting of model architecture and diagnostic thresholds [14,41]. A lack of interpretability in many deep learning models can hinder clinician acceptance. Emerging solutions include federated learning to preserve privacy across institutions, and explainable AI tools that increase model transparency and trustworthiness [43,44,45].

Finally, ethical considerations are gaining prominence. AI models trained on imbalanced data may perpetuate healthcare disparities [46]. Future systems should incorporate representative training datasets, conduct bias auditing, and apply standardized reporting frameworks such as STARD-AI and TRIPOD-AI to ensure transparency and fairness [41,42]. These actions will be critical to supporting the responsible and equitable deployment of AI in TSCC diagnosis.

### 4.4. Study Limitations

Several limitations of this systematic review should be acknowledged. First, the number of included studies was relatively small (*n* = 13), and the heterogeneity in study design, patient populations, imaging modalities, and AI algorithms limited the possibility of subgroup or meta-regression analyses. Second, most studies relied on internal validation strategies such as k-fold cross-validation or split-sample testing, with only one study conducting external validation [22]. This limitation highlights concerns regarding the generalizability and external validity of the models. Third, selection bias and unclear reporting were observed in many studies. According to the QUADAS-2 assessment, less than half of the included studies were rated as low risk across key domains [14]. Fourth, the review selected only the best-performing AI model per study for meta-analysis, which could introduce a best-case scenario bias. To mitigate this, we conducted a sensitivity analysis using the worst-performing models, which confirmed a statistically significant performance decline yet supported the overall validity of the pooled findings (Table 2). Specifically, the average sensitivity decreased by 22.6 percentage points (from 0.815 to 0.589, *p* = 0.0026), and specificity declined by 9.2 percentage points (from 0.838 to 0.746, *p* = 0.0526), indicating a statistically significant overestimation of sensitivity when only best-performing models are included. Finally, the review included only studies published after 2020 and written in English, potentially introducing time frame and language bias.

## 5. Conclusions

This systematic review underscores the growing role of AI in the comprehensive management of TSCC, encompassing diagnosis, prognostication, treatment planning, and post-treatment surveillance. Across the 13 included studies, AI models—particularly those utilizing deep learning architectures such as CNNs—demonstrated strong diagnostic performance, with pooled sensitivity and specificity values exceeding those typically achieved by conventional imaging modalities. Multimodal approaches that integrate radiologic, histopathologic, and clinical data further enhanced predictive accuracy and clinical applicability (Table 1).

Beyond diagnostic tasks, AI has shown increasing utility in prognostic modeling, tumor microenvironment analysis, and surgical decision support. Recent integrative frameworks incorporating omics-based signatures, immune profiling, and liquid biopsy data represent a promising frontier for personalized care strategies in TSCC. Concurrently, technological innovations such as AR-assisted surgery, explainable AI (XAI), and federated learning are fostering the development of systems that are not only accurate but also interpretable, privacy-preserving, and scalable across institutions.

However, several limitations remain. Methodological heterogeneity, limited external validation, and reliance on small, demographically homogeneous datasets hinder generalizability and clinical translation. Many studies continue to lack adherence to standardized reporting frameworks such as STARD-AI and TRIPOD-AI, impeding reproducibility and cross-study comparisons. Ethical concerns, including algorithmic bias and unequal performance across demographic subgroups, must also be proactively addressed to ensure equitable implementation in diverse patient populations.

In conclusion, while AI is not a substitute for clinical expertise, it is rapidly evolving into a transformative adjunct within multidisciplinary TSCC management. To unlock its full potential, future research must prioritize robust external validation, multi-institutional collaboration, and the development of transparent, ethically aligned, and clinically integrated AI solutions for head and neck oncology.

## Figures and Tables

**Figure 1 biomedicines-13-01849-f001:**
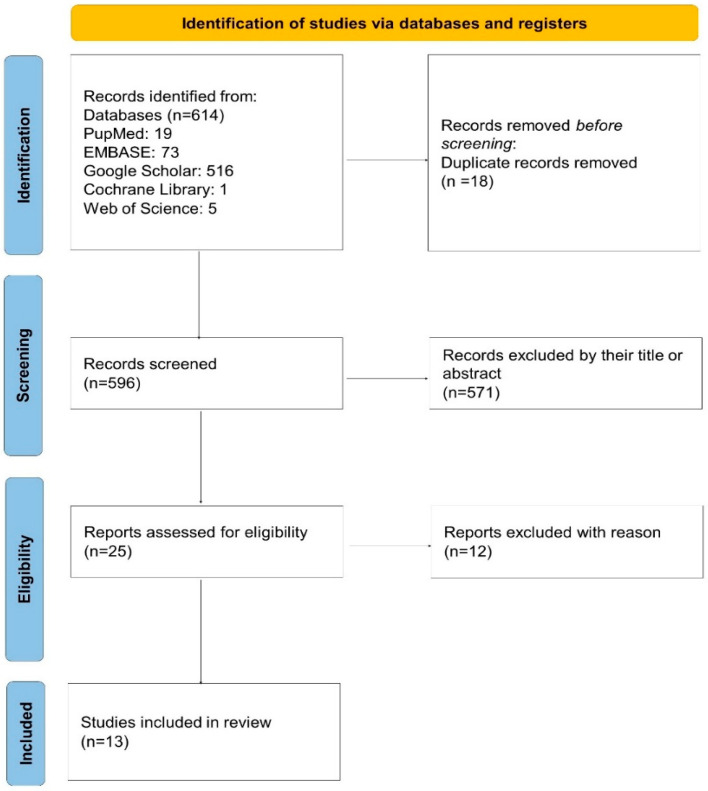
PRISMA 2020 flow diagram of the study selection process. A total of 614 records were identified, 596 were screened after removing duplicates, and 25 full-text articles were assessed for eligibility. Thirteen studies were finally included in the review.

**Figure 2 biomedicines-13-01849-f002:**
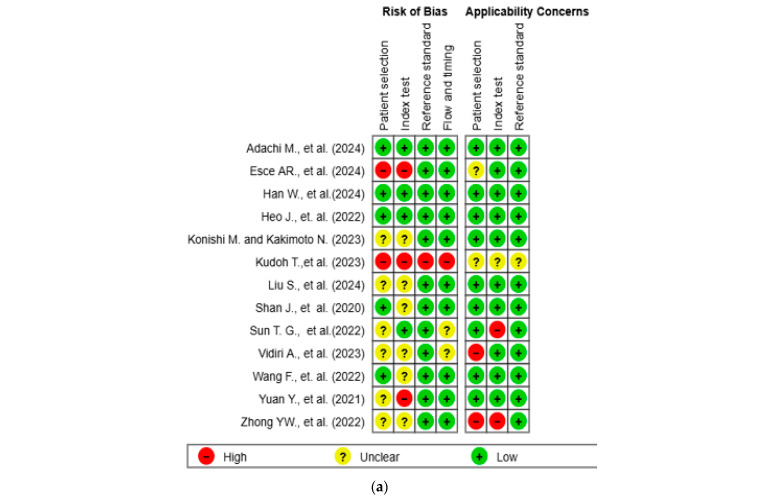
Methodological quality assessment of the included studies using the QUADAS-2 tool. (**a**) Risk of bias and applicability concerns summary for individual studies. Each row represents a single study, and each column represents one of the four QUADAS-2 domains—patient selection, index test, reference standard, and flow and timing—for both risk of bias (**left**) and applicability concerns (**right**). Color codes indicate judgments: green for low risk, yellow for unclear, and red for high risk. (**b**) Aggregate summary graph illustrating the proportion of studies rated as low, high, or unclear risk of bias and applicability concerns in each QUADAS-2 domain. Notably, the highest proportions of high and unclear risk were observed in the domains of patient selection and index test, while reference standard and flow and timing were predominantly low-risk [10,15,16,17,18,19,20,21,22,23,24,25,26].

**Figure 3 biomedicines-13-01849-f003:**
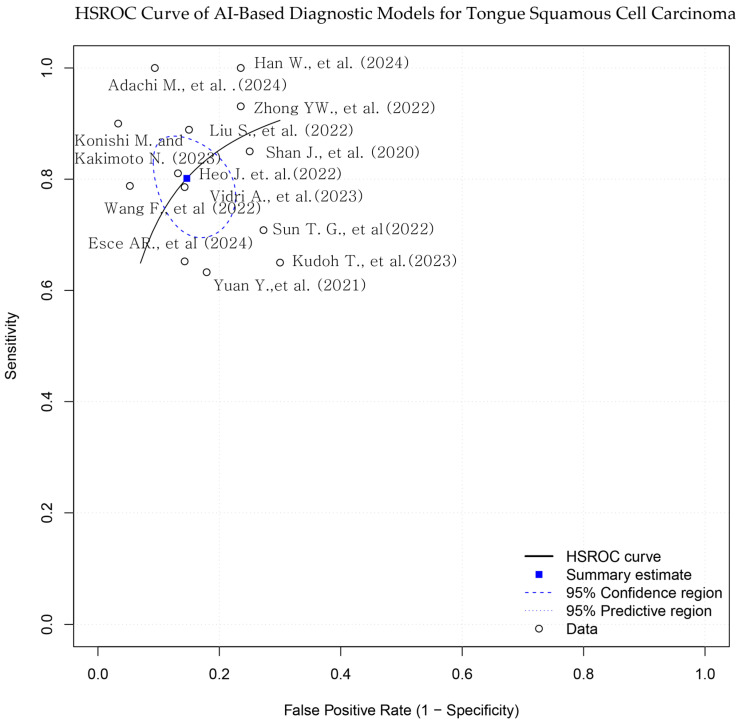
Hierarchical summary receiver operating characteristic (HSROC) curve for AI-based models in tongue squamous cell carcinoma (TSCC) diagnosis. The blue square represents the summary point for pooled sensitivity and specificity, derived from a bivariate random-effects model. Each black dot indicates an individual study. The dashed ellipse shows the 95% confidence region for the summary point, and the solid ellipse indicates the 95% prediction region. The diagonal dashed line represents the line of no discrimination [10,15,16,17,18,19,20,21,22,23,24,25,26].

**Figure 4 biomedicines-13-01849-f004:**
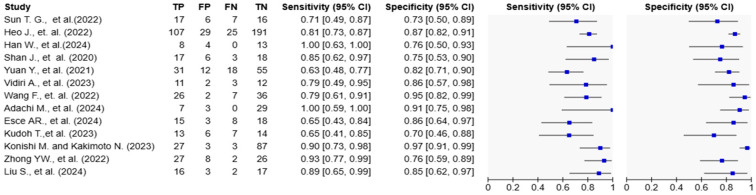
Forest plots of diagnostic accuracy metrics for AI-based models in tongue cancer. The table on the left displays individual study data, including true positives (TP), false positives (FP), false negatives (FN), and true negatives (TN), along with calculated sensitivity and specificity with 95% confidence intervals (CIs). The plots on the right visually represent the sensitivity and specificity estimates for each study, with blue squares indicating point estimates and horizontal lines representing the 95% CIs. The variation in confidence interval widths reflects differences in sample size and precision across studies [10,15,16,17,18,19,20,21,22,23,24,25,26].

**Table 1 biomedicines-13-01849-t001:** Summary of AI-based diagnostic studies for tongue cancer.

Author	Year	Modality	AI Model	Data Type	Sample Size	Validation	AUC	Sensitivity (%)	Specificity (%)
Han W, et al. [10]	2024	Contrast-enhanced CT	Integrated (Radiomics + DL + Clinical, Stacked Ensemble)	Radiomics	125	Internal validation: Stratified 5-fold cross-validation	0.949	100.0	76.5
Yuan Y, Ren J, Tao X. [15]	2021	MRI (T2WI + ceT1WI)	NB	Radiomics texture features	116	10-fold cross-validation	0.802	63.3	82.1
Wang F, et al. [16]	2022	MRI (T2-weighted)	SVM	Radiomics features + Clinicopathological data	236	Internal validation using 5-fold cross-validation	0.872	78.78	93.47
Konishi M, Kakimoto N. [17]	2023	Intraoral ultrasonography (US)	NTB	Hypoechoic region + 3 mm margin ROI, Pyradiomics, 15/850 by LASSO	120	internal validation + 5-fold cross-validation	0.967	90	96.7
Kudoh T, et al. [18]	2023	18F-FDG PET	LASSO-based radiomics model	Radiomics	40	5-fold cross-validation	0.79	65	70
Vidiri A, et al. [19]	2023	MRI (post-contrast T1-weighted high-resolution imaging)	NB classifier	MRI-based DOI + tumor dimensions + shape-based + intensity-based radiomics features	108	Internal Split: 80 (training), 28 (validation/test)	0.81	76.5	70.8
Liu S, et al. [20]	2024	MRI (multiple sequences including T1WI, FS-T2WI, T2WI, CE-MRI, ADC)	Multimodal MRI radiomics (T1WI, FS-T2WI, T2WI, CE-MRI)	Multimodal MRI	400	Internal validation (split training: test = 7:3)	0.822	86.0	80.0
Zhong YW, et al. [21]	2022	CT (Radiomics + clinical LN status)	ANN	CT-based tumor radiomics + clinical LN info	313	Internal split (Train: 60%, Validation: 20%, Test: 20%)	0.943	93.1	76.5
Heo J, et al. [22]	2022	Oral endoscopic images	CNN –DenseNet169	Raw endoscopic images (no segmentation), clinical cases	5579 endoscopic images	External test set (from 5th institution)	0.895	79.3	85.3
Sun TG, et al. [23]	2022	Contrast-enhanced CT (CECT)	Inception-ResNet-V2 (CNN)	Medical images (180 × 180 × 3 PNGs from CECT)	179 patients/4510 CT images	Train/Validation/Test split (hold-out validation)	0.717	70.8	72.7
Shan J, et al. [24]	2020	Clinical + histopathology	RF	Tumor size, DOI, differentiation	145	Hold out (70/30 split) + Stratified K-Fold cross-validation and GridSearchCV.	0.786	85	75
Adachi M, et al. [25]	2024	WSIs, HE-stained pathology + clinicopathological data	CLAM + SVM (WSI + clinicopathologic fusion)	WSI-based AI-extracted features	148	CLAM (Attention-based MIL + ResNet feature extractor)	0.991	100	90.6
Esce AR, et al. [26]	2024	Histopathology (H&E-stained whole-slide images)	HALO-AI (CNN, VGG)	Image patches from tumor regions (no clinical or radiomic data)	108 images (from 89 patients)	Internal split into train/test sets	0.729	65.0	86.0

Abbreviations: AUC: Area Under the Curve; ADC: Apparent Diffusion Coefficient; ANN: Artificial Neural Network; CE-MRI: Contrast-Enhanced Magnetic Resonance Imaging; CECT: Contrast-Enhanced Computed Tomography; CNN: Convolutional Neural Network; DL: Deep Learning; DOI: Depth of Invasion; FS-T2WI: Fat-Suppressed T2-Weighted Imaging; H&E: Hematoxylin and Eosin; MIL: Multiple Instance Learning; NB: Naïve Bayes; NTB: Neural Tanh Boost; PET: Positron Emission Tomography; ROI: Region of Interest; SVM: Support Vector Machine; T1WI: T1-Weighted Imaging; T2WI: T2-Weighted Imaging; US: Ultrasonography; VGG: Visual Geometry Group; WSI: Whole-Slide Image.

**Table 2 biomedicines-13-01849-t002:** Sensitivity and specificity comparison between best- and worst-performing AI models included in the meta-analysis.

Model Selection	Average Sensitivity	Average Specificity
Best-performing models	0.815	0.838
Worst-performing models	0.589	0.746
Mean difference	−0.226 (*p* = 0.0026)	−0.092 (*p* = 0.0526)

Footnote: This sensitivity analysis was conducted to assess potential bias introduced by selecting only the highest-performing model per study. The results demonstrate a statistically significant overestimation of sensitivity and a marginal reduction in specificity, suggesting the importance of including multiple model performances in future meta-analyses.

## Data Availability

No new data were created or analyzed in this study. Data sharing is not applicable to this article.

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
