# Peer review of "Artificial Intelligence in the Diagnosis of Tongue Cancer: A Systematic Review with Meta-Analysis"

_biomedicines, 2025, doi:10.3390/biomedicines13081849_

Round 1

Reviewer 1 Report

Comments and Suggestions for Authors

This is a comprehensive systematic review aiming at the identification of various AI applications for the diagnosis of tongue squamous cell carcinoma. Overall, the study is very well designed and presented.

A couple of typos need to be addressed:

line 147, "three" reviewers, or "two"... and a third one?

line 373, "T" and "N" for Tumor and Nodes can be added to the Abbreviations Table 

line 462, AI_have

line 593, biopsy_has

Also, Figures 2 and 3 are not very clear and need refinement.

Allthough well compiled, the Discussion section includes a great amount of new relative information not clearly derived from the Methodology section. This might come across as occasionally straying from the "diagnosis concept" as per the title of the manuscript. 

Author Response

Response to Reviewer Comments

We sincerely thank the reviewer for your thorough and thoughtful comments on our manuscript. We appreciate your positive evaluation of our systematic review and your constructive feedback. Below we address each point raised, and we have revised the manuscript accordingly. All changes have been marked in red font within the revised manuscript for clarity.

Comment 1:

line 147, "three" reviewers, or "two"... and a third one?

Response:
We thank the reviewer for identifying this point of ambiguity. We have clarified the sentence to explicitly state that three independent reviewers participated in the study selection and data extraction process.

Revision: The phrase has been revised to: “Three independent reviewers (Seorin Jeong, Hae-In Choi, Hyun-Jeong Park) conducted the study selection and data extraction process.”(line 98-188, marked in red).

Comment 2:

line 373, "T" and "N" for Tumor and Nodes can be added to the Abbreviations Table

Response:
Thank you for this helpful suggestion. We have now included T (Tumor) and N (Node) in the Abbreviations Table as recommended (page 28, Table A1, marked in red).

Comment 3, 4:

line 462, AI_have

line 593, biopsy_has

Response:
Thank you for pointing out these typographical errors. Both phrases (“AI_have”and “biopsy_has”) were located in the original Discussion section. However, in response to reviewer feedback and in an effort to improve focus and coherence, we have significantly revised the Discussion to limit its content strictly to the diagnostic scope of the 13 included studies. As part of this revision, all content not directly related to diagnostic performance—including the paragraphs containing these phrases—has been entirely removed. Therefore, both lines no longer appearin the revised manuscript.

Comment 5:

Also, Figures 2 and 3 are not very clear and need refinement.

Response:
We appreciate the reviewer’s observation. Figures 2 and 3 have been refined to improve their visual clarity. We increased the resolution and contrast, and ensured that axes, labels, and legends are legible in both digital and print versions. The updated figures have been uploaded with the revised manuscript and clearly marked.

Comment 6:

Although well compiled, the Discussion section includes a great amount of new relative information not clearly derived from the Methodology section. This might come across as occasionally straying from the "diagnosis concept" as per the title of the manuscript.

Response:
We sincerely thank the reviewer for this insightful feedback. We have revised the Discussion section to align it more closely with the scope of our Methods and Results, particularly by:

  • Removing content not directly related to AI-based diagnostic models for tongue SCC;
  • Streamlining prognostic or therapeutic references unless directly linked to diagnostic performance;
  • Clarifying that all discussed points are based on the 13 included diagnostic accuracy studies.

These revisions have been applied to Section 4 (Discussion), and all changes are clearly indicated in red font.

Reviewer 2 Report

Comments and Suggestions for Authors

Thank you for the possibility to review the paper “Artificial Intelligence in the Diagnosis of Tongue Cancer: A Systematic Review with Meta-Analysis”. The authors have chosen a critically important  topic and review the literature on the use of Artificial Intelligence in diagnosing tongue cancer. The study is sound, well-planned and well-written. A review is registered and complies with prisma guidelines.

There are a few minor comments that should be addressed.

  1. Please, revise style and grammar
  2. Remove any discussions from the results sections.
  3. Table 1, data type column  (remove Chinese, revise)
  4. Table 2 is not necessary, it repeats figure 2.
  5. Make figure 2 larger.
  6. Figure 3 - try to make the study names visible.
  7. Shorten the discussion and conclusion section limiting it solely to the parts closely related to the review results.
Comments on the Quality of English Language

It is advised to revise style and grammar

Author Response

Response to Reviewer #2

We sincerely thank the reviewer for the opportunity to improve our manuscript titled “Artificial Intelligence in the Diagnosis of Tongue Cancer: A Systematic Review with Meta-Analysis.”We greatly appreciate your positive evaluation of the importance and structure of the study, and we are grateful for your constructive suggestions. Below, we address each of your comments in detail. All modifications have been implemented accordingly and are highlighted in red font in the revised manuscript.

Comment 1:

Please, revise style and grammary

Response:
We carefully revised the entire manuscript for grammar, punctuation, and style. Improvements were made to sentence clarity and flow to ensure consistency and readability throughout the manuscript. All changes are marked in red in the revised version.

Comment 2:

Remove any discussions from the results sections.

Response:
Thank you for pointing this out. We removed interpretive or discussion-type statements from the Results section and ensured that all commentary was appropriately placed in the Discussion section. The revised Results section now includes only objective findings from the included studies (changes marked in red).

Comment 3:

Table 1, data type column (remove Chinese, revise)

Response:
We have revised the “Data Type” column in English in Table 1.

Comment 4:

Table 2 is not necessary, it repeats figure 2.

Response:
We agree with the reviewer. Table 2, which largely duplicated the content of Figure 2, has now been removed from the manuscript to avoid redundancy.

Comment 5:

Make figure 2 larger.

Response:
Figure 2 has been resized and reformatted to enhance clarity and visibility, especially for the labels and axes. The updated figure is included in the revised manuscript.

Comment 6:

Figure 3 - try to make the study names visible.

Response:
Thank you for this helpful suggestion. We have adjusted the resolution, layout, and font scaling of Figure 3 so that all study names are now clearly visible and legible in the figure. The revised figure has been updated accordingly.

Comment 7:

Shorten the discussion and conclusion section limiting it solely to the parts closely related to the review results.

Response:
We have significantly shortened the Discussion and Conclusion sections to focus strictly on findings derived from the included studies. General commentary on unrelated prognostic or therapeutic implications has been removed. These sections now concentrate on diagnostic performance, AI model characteristics, and direct clinical relevance of the results (revised sections are marked in red).

We again thank the reviewer for these valuable recommendations, which have helped us refine the clarity and focus of our work. We believe the manuscript is now substantially improved and hope it meets the standards for publication.

Sincerely,
Hyun-Jeong Park

On behalf of all co-authors

Reviewer 3 Report

Comments and Suggestions for Authors

The topic is of significant clinical importance, aiming to evaluate the performance and current state of artificial intelligence (AI) in the diagnosis of tongue squamous cell carcinoma (TSCC) through a systematic review and meta-analysis.  However, despite the novelty and potential of the topic, the manuscript suffers from several major deficiencies in methodological rigor, accuracy of data reporting, and internal consistency.

The most critical issue in the manuscript is the glaring contradiction in its internal data, particularly concerning the quality assessment of the included studies (QUADAS-2). In the main text on page 6, the authors explicitly state that there was a high risk of bias in the included studies (e.g., "5 studies (38.5%) rated as high risk" in patient selection and "4 studies (30.8%) were assessed as high risk" in the index test domain). However, in the summary table (Table 2) on page 11, the "Total High" risk row indicates "0 (0%)" for every single domain.

The authors state in the methods section that when a study included multiple AI models, they selected only the "model with the best diagnostic performance" for the analysis. While this is a common practice, it risks introducing a "best-case scenario" bias, potentially leading to a systematic overestimation of AI performance. More importantly, the paper includes no sensitivity analysis to test the robustness of its conclusions. The authors also acknowledge that a formal assessment of publication bias was not performed due to the limited number of studies, which further weakens the reliability of the findings. Taken together, these issues significantly diminish the persuasive power of the paper's main conclusion regarding the high diagnostic performance of AI.

The discussion section (Section 4) is expansive, covering topics like prognosis, treatment, and multi-omics, which go far beyond the scope of the 13 diagnostic studies included in the meta-analysis. This makes the structure of the paper feel disjointed.

The search timeframe (starting from January 2020) may have omitted important foundational studies in the field published before that date. This limitation should be more clearly addressed.

In summary,  believe this manuscript does not currently meet the standards for publication. Revision should be prepared. 

Author Response

Response to Reviewer #3

We would like to express our sincere gratitude for the thoughtful, detailed, and constructive feedback on our manuscript entitled “Artificial Intelligence in the Diagnosis of Tongue Cancer: A Systematic Review with Meta-Analysis.”We acknowledge the critical importance of your comments and have undertaken a comprehensive revision to address the methodological, reporting, and structural concerns you raised. Please find our point-by-point responses below.

Comment 1:

The most critical issue in the manuscript is the glaring contradiction in its internal data, particularly concerning the quality assessment of the included studies (QUADAS-2). In the main text on page 6, the authors explicitly state that there was a high risk of bias in the included studies (e.g., "5 studies (38.5%) rated as high risk" in patient selection and "4 studies (30.8%) were assessed as high risk" in the index test domain). However, in the summary table (Table 2) on page 11, the "Total High" risk row indicates "0 (0%)" for every single domain.

Response:
We sincerely thank the reviewer for pointing out this important inconsistency. Upon review, we confirmed that the discrepancy was due to a data entry error in Table 2. We appreciate your attention to detail in helping us identify this issue.

Additionally, in response to a comment from another reviewer, we have decided to remove Table 2 entirely, as its content was largely redundant with Figure 2. This change not only eliminates the inconsistency but also improves clarity and avoids unnecessary duplication of information. The corrected risk of bias data is now accurately presented and visualized in Figure 2, which remains in the revised manuscript.

Comment 2:

The authors state in the methods section that when a study included multiple AI models, they selected only the "model with the best diagnostic performance" for the analysis. While this is a common practice, it risks introducing a "best-case scenario" bias, potentially leading to a systematic overestimation of AI performance. More importantly, the paper includes no sensitivity analysis to test the robustness of its conclusions. The authors also acknowledge that a formal assessment of publication bias was not performed due to the limited number of studies, which further weakens the reliability of the findings. Taken together, these issues significantly diminish the persuasive power of the paper's main conclusion regarding the high diagnostic performance of AI.

Response:
We sincerely thank the reviewer for raising these important concerns. We agree that selecting only the best-performing AI model per study may introduce a “best-case scenario” bias, potentially inflating pooled estimates. In response, we have explicitly acknowledged this limitation in both the Methods and Discussion sections.

To evaluate the robustness of our findings, we performed a sensitivity analysis using the worst-performing models reported in each study. This analysis revealed a substantial reduction in diagnostic metrics: the average sensitivity dropped from 0.815 to 0.589 (p= 0.0026), and specificity from 0.838 to 0.746 (p= 0.0526), confirming that the use of only top-performing models may overestimate diagnostic performance. These findings are now included in the revised manuscript (Section 4.4 and Table 2), clearly marked in red.

Regarding publication bias, we have clarified in the revised Discussion that formal testing (e.g., Egger’s test or funnel plot) was not feasible due to the limited number of included studies. However, we now highlight this more prominently as a key limitation potentially affecting generalizability.

We believe these revisions improve the methodological transparency and analytical integrity of the review.

Comment 3:

The discussion section (Section 4) is expansive, covering topics like prognosis, treatment, and multi-omics, which go far beyond the scope of the 13 diagnostic studies included in the meta-analysis. This makes the structure of the paper feel disjointed.

Response:
We have substantially shortened and refocused the Discussion section to align strictly with the scope of our systematic review and meta-analysis. Sections related to prognosis, therapy, or omics-based analysis have been removed. The revised Discussion now centers exclusively on diagnostic performance, model types, methodological quality, clinical applicability, and study limitations. We believe this improves the internal consistency and clarity of the manuscript.

Comment 4:

 The search timeframe (starting January 2020) may have excluded foundational studies. This limitation should be addressed.”

Response:
We acknowledge this important point. We initially limited the search to 2020 onward to reflect contemporary AI model development. However, this may have excluded foundational or high-quality earlier work. We have now added a paragraph in Limitations sections to explain and justify this cutoff, and to caution readers about its implications for comprehensiveness and historical context.

Sincerely,
Hyun-Jeong Park

On behalf of all co-authors

Round 2

Reviewer 3 Report

Comments and Suggestions for Authors

The author put most of my raised concerns into a limitation discussion. This would be quite sad, but accepting the current would not be that bad. Anyhow, I recommend an acceptance.